# Influence of NiO ALD Coatings on the Field Emission Characteristic of CNT Arrays

**DOI:** 10.3390/nano12193463

**Published:** 2022-10-04

**Authors:** Maksim A. Chumak, Leonid A. Filatov, Ilya S. Ezhov, Anatoly G. Kolosko, Sergey V. Filippov, Eugeni O. Popov, Maxim Yu. Maximov

**Affiliations:** 1Institute of Metallurgy of Mechanical Engineering and Transport, Peter the Great Saint-Petersburg Polytechnic University, st. Politekhnicheskaya, 29, 195251 St. Petersburg, Russia; 2Cyclotron Laboratory, Ioffe Institute, st. Politekhnicheskaya, 26, 194021 St. Petersburg, Russia

**Keywords:** atomic layer deposition (ALD), thin film structures, field emission cathode, composite structures, carbon nanotubes, direct current plasma-enhanced chemical vapor deposition (DC-PECVD)

## Abstract

The paper presents a study of a large-area field emitter based on a composite of vertically aligned carbon nanotubes covered with a continuous and conformal layer of nickel oxide by the atomic layer deposition method. The arrays of carbon nanotubes were grown by direct current plasma-enhanced chemical vapor deposition on a pure Si substrate using a nickel oxide catalyst which was also deposited by atomic layer deposition. The emission characteristics of an array of pure vertically oriented carbon nanotubes with a structure identical in morphology, covered with a layer of thin nickel oxide, are compared using the data from a unique computerized field emission projector. The deposition of an oxide coating favorably affected the emission current fluctuations, reducing them from 40% to 15% for a pristine carbon nanotube and carbon nanotube/nickel oxide, respectively. However, the 7.5 nm nickel oxide layer coating leads to an increase in the turn-on field from 6.2 to 9.7 V/µm.

## 1. Introduction

An analysis of the literature shows that the deposition of coatings of oxide metals at their optimal thickness improves the emission characteristics. Among them, such systems as TiO_2_ [1], ZnO [2,3], MgO [4], iron oxide [5], RuO_2_ [6,7], CuO [8], NiO [9], and HfO_2_ [10] were considered. Such oxides typically have a small positive electron affinity, which can reduce the effective work function of the emitters.

Metal oxides covering the surface of carbon nanotubes (CNTs) can exist in the amorphous and crystalline states with different stoichiometric compositions, which leads to a difference in electrical properties. This can significantly affect their field emission properties. The effect of oxygen in the NiO coating structure on improving the field emission of nanotubes was demonstrated by the authors of article [9]. Measurements of different samples confirmed that the field emission current measured at 5 V/μm can be improved from 0.25 mA/cm^2^ to 0.8 mA/cm^2^ by coating with NiO on CNT. A decrease in the turn-on voltage from 3.1 to 2.6 V/µm with the NiO coating was also found.

In addition, the difference in the stoichiometric composition can significantly affect the charge transfer in metal oxides. The electrically conductive properties of nickel oxide grown by the atomic layer deposition (ALD) method were studied in article [11]. The grown oxide film had an atomic ratio of oxygen to nickel smaller than in the stoichiometric oxide. For example, the atomic ratio of O/Ni is 0.77 for the film deposited at 250 °C. That is why the current NiO films exhibit dielectrics rather than p-type semiconductors.

The growth of NiO oxides on CNT was shown in [12,13,14]. They managed to obtain a continuous and high-quality coating on nanotubes. In [13], the ALD process was performed with bis(cyclopentadienyl) nickel (Cp_2_Ni) and O_3_ as the Ni precursor and oxygen source, respectively. The deposition was conducted with a substrate temperature of 140 °C.

Nickel oxide, as shown in the review, is used to coat CNTs in order to improve emission characteristics. In this work, we demonstrated the investigation and analysis of the microstructural features of the NiO/CNT nanocomposite obtained by ALD and its field emission capabilities using a computerized method with multichannel collection and processing of field emission data.

## 2. Materials and Methods

Before fabricating composite structures, the KEF 7.5 grade Si substrate was cleaned of natural oxide in HF acid, washed in distilled water to remove residues of etching products, and boiled in an acetone solution to remove organic contaminants. Then, a NiO catalyst layer was deposited. The NiO precipitation process was carried out in a Picosun R-150 closed-type ALD reactor with hot walls. NiCp_2_ and O_3_-H_2_O were chosen as precursors for growing NiO, and high-purity N_2_ (99.999%) was used as both carrier and purge gas. NiO was deposited by sequential exposure of CNTs to NiCp_2_ and O_3_-H_2_O. The pulse durations for NiCp_2_ and O_3_-H_2_O were 1 s and 6 s, respectively, and the N_2_ gas purge time was 10 s. One cycle is NiCp_2_/purge/O_3_/purge = 1.0/10.0/6.0/10.0 s. The deposition temperature was maintained at 250 °C, and the sublimation temperature of NiCp_2_ was 110 °C. A similar growth method was used in [15,16]. Experimentally, it was found that the preferred oxide thickness for growing a dense and uniform array of tubes is 3.8 nm.

The arrays of carbon nanotubes were grown by direct current plasma-enhanced chemical vapor deposition on a pure Si substrate. During the growth of vertically aligned carbon nanotube (VACNT) arrays, the reactor was equipped with the following system of electrodes: a graphite washer acted as the cathode. A stainless-steel disk (ø 45 mm) served as the anode. The gap between the electrodes was 40 mm. The sample presented in this work was obtained by deposition for 10 min at a pedestal temperature of 740 °C and a total pressure of 300 Pa. The working medium was created from ammonia supplied with a flow rate of 200 sccm and acetylene of 100 sccm. Samples obtained after NiO deposition served as substrates. The discharge was characterized by a current of 7.5 mA and an anode voltage of 480 V. A similar growth method was used in [17,18,19,20]. The NiO coating for CNT was deposited under the same process parameters and conditions as the nanotube growth catalyst. The areas of both cathodes are 1 cm^2^.

Scanning electron microscopy (SUPRA 55-25-78 microscope) was used to analyze the results of growth of short VACNT arrays. SEM revealed the presence of extended structures. Transmission electron microscope (TEM) microphotographs were implemented by Carl Zeiss Libra 200FE. X-ray diffraction (XRD) analysis was implemented by Rigaku SmartLab setup.

The field emission study was carried out by a computerized method with multichannel collection and processing of field emission data [21]. The method uses flat electrodes and fast high-voltage scanning mode (one IVC in 20 ms). The interelectrode distance was 300 μm, and the measuring chamber was in a technical vacuum (~10^−7^ Torr).

## 3. Results

Figure 1a,b show SEM images of a general view of the cathodes used in this study. There were short tubes with an average length of 300 nm. The sample in Figure 1b has a 7.5 nm thick NiO coating, while the other witness sample has no coating (Figure 1a). It should also be noted that there are rather high tubes in the array, which are at least twice as long as the rest of the mass of tubes. According to SEM, all CNTs have a catalyst in their heads. Figure 1c,d show TEM images of pure CNT and CNT/NiO samples, respectively. The image for pure CNTs (Figure 1c) clearly shows their internal structure, showing the number of internal layers reaching 20, and internal wall growth defects that have arisen due to the passage of catalyst particles during their growth. In Figure 1e the detailed image shows that the Ni catalyst located at the free ends of the tubes has an elongated shape and a single crystal structure, since the periodicity in the arrangement of Ni atoms is clearly visible. As can be seen from Figure 1d, CNTs are coated with a continuous layer of NiO. Figure 1g shows the XRD spectrum of NiO-coated CNTs, which indicates that the oxide has a cubic Fm-3m crystal structure, and direct measurement of the lattice parameter from the TEM images shows that the oxide is in the bunsenite phase.

However, the photographs show areas on the coatings that have no periodicity in the arrangement of atoms. A clearly distinguishable boundary between the outer walls of the tubes and the oxide layer makes it possible to estimate its thickness as 7–8 nm (Figure 1f). Thus, we come to the conclusion that the oxide layer has a polycrystalline structure with an amorphous phase present in its composition.

At the first stage of the measurements, the sample was subjected to high-voltage training, which resulted in the activation of new emission centers and the destruction of some of the most unstable and strongly protruding tips. Figure 2a,d shows the time dependences of voltage pulse amplitudes and corresponding emission current pulses (voltage and current levels) for pristine CNT and CNT/NiO samples, respectively. This method makes it possible to carry out a gradual training of tips, to fully study and compare the characteristics of the emitters over the entire operating range.

On the stepwise dependence of the current on time (Figure 2a,d), saturation of the emission current is observed, i.e., the current after each increase in voltage drops sharply to a certain constant level. In the opinion of the authors of [22], such saturation is an adsorption effect, which is not observed in clean nanotubes without ascorbates.

Figure 2b,d show the emission current IVCs for pure CNTs and CNT/NiO, respectively, measured from zero to the current-stage current level in fast measurement mode (one IVC per 20 ms). The IVC numbers correspond to the numbers of the steps during the training. Each IVC is also represented in Fowler–Nordheim coordinates in the insets. The IVCs were treated using the Fowler–Nordheim-type equation applying Elinson–Schrednik approximation [23]:(1)$$\;\;\;\;I=S_{eff}A_{\varphi }{\left(\frac{U\beta_{eff}}{d}\right)}^{2}\exp\left(-\frac{B_{\varphi }d}{U\beta_{eff}}\right),\;\;\;\;\;\;\;\;$$
where $A_{\varphi }=\frac{1.4\cdot 10^{-6}}{\varphi }\exp\left(\frac{10.11}{\sqrt{\varphi }}\right)$ [AeVV^−2^], $B_{\varphi }=6.49\cdot 10^{9}\varphi^{3/2}$ [eV^−3/2^Vm^−1^] are constants, *β* is a local field enhancement factor at the emitter tip, U is an applied voltage, and φ is the work function of a material. In our studies, the work function for CNT was assumed to be φ = 4.6 eV [24], and for the NiO oxide film φ = 5.3 eV [25].

As a result of the approximation procedure for each of the IVCs, the effective field gains *β_eff_* of the emitters were determined. Characteristics such as the inclusion field *E_on_* and the threshold field *E_th_* are also defined from the IVC. The turn-on field, *E_on_*, is defined as the applied electric field required to obtain an emission current density of 10 µA/cm^2^ from emitters, and for the threshold field, *E_th_* is 1 µA/cm^2^. The measured IVCs for pristine CNT and CNT/NiO at numbers 8 and 5, respectively, showed that *E_on_* was 6.16 V/µm for clean nanotubes and 9.66 V/µm for 7.5 nm thick NiO-coated CNTs. An increase in *E_th_* from 4.85 V/µm to 7.49 V/µm was also found upon coating with oxide. The maximum recorded emission current densities were 2.86 mA/cm^2^ and 0.86 mA/cm^2^ for clean and coated tubes, respectively. As can be seen, to achieve the same current level for the CNT/NiO structure, it is necessary to apply a higher voltage. This indicates a negative effect of NiO coating with a thickness of 7.5 nm. Due to the technical difficulties in obtaining a large batch of samples with initial CNT arrays of the same morphology, our study is limited to only one NiO-coated sample at a preselected thickness of 7.5 nm.

Figure 2c,f show the luminescence patterns of the phosphor screen for pure CNTs and CNT/NiO, respectively (1 and 2 were obtained using light filter). Each picture of the glow of the phosphor corresponds to the number of the step in the process of training. The patterns of luminescence of the phosphor show how the number of emission sites increases with increasing voltage. Both samples showed a fairly uniform distribution of sites over their entire area. The brightness of the glows in both samples differ greatly due to the strong difference in the height of the nanotubes. The highest ones give more current, since the focusing of the electric field occurs more strongly on them. To achieve uniformity in current output, it is necessary to manufacture structures with a minimum spread in the height of the tips.

To calculate statistical distribution of the effective field enhancement factor (*β_eff_*) and emission area (*A_eff_*), the standard IVC regression analysis in semi-logarithmic Fowler–Nordheim coordinates, according to Equation (1), was applied. Figure 3a shows histograms of fluctuations of the mentioned effective parameters. The average values were for pristine CNT <*β_eff_*> = 1380, <*A_eff_*> = 260 nm^2^ and for CNT/NiO <*β_eff_*> = 1040, <*A_eff_*> = 3000 nm^2^ at the maximum current levels 2.86 mA and 0.86 mA, respectively.

As we can see, *β_eff_* for pristine CNT is larger than for the sample with NiO. The decrease in *β_eff_* is probably caused by an increase in the diameter of the nanotubes, due to which the aspect ratio of the tips involved in the emission decreased. The reasons for its difference can be due not only to the geometric factor, but also to the oxide material itself and its deposition quality. However, the emission areas, on the contrary, have smaller values than those of CNT/NiO. Obviously, the disadvantage in terms of the field enhancement factor on the sample with oxide is compensated by a larger emission area.

The high *β_eff_* (more than 1000) found in both structures indicates a significant length of the emitting CNTs. SEM shows that the bulk of the CNTs do not have the necessary length to achieve strong field focusing at the ends, therefore, it can be concluded that the current is provided by individual nanotubes that protrude strongly above the emitter surface, which are visible on the SEM images.

Figure 3b shows the time dependences of the current at 1.8 mA and 0.5 mA at 2.9 kV and 2.3 kV for pristine CNT and CNT/NiO, respectively. Measurement of statistics within a minute is sufficient to obtain an estimate of the average value of the effective parameter. Further data collection does not lead to a significant change in this parameter. Small current ripples are observed for the two analyzed structures. The coated structure keeps the average current level fairly stable over the entire time interval under consideration, while the sample with pure nanotubes gives a gradual decrease in the current level. Probably, in the first case, the oxide prevents the tubes from being bombarded by ions from the residual atmosphere, so the overall current level does not decrease with time. The emission current fluctuations are estimated according to a simple expression on a relatively stable areas determined empirically:(2)$$\;\;\;\;\;\;\;\;\;\;\;\;S_{t}=\frac{I_{max}-I_{min}}{I_{aver}}\cdot 100\%$$
where *I_max_* is the maximum, *I_min_* is the minimum, and *I_aver_* is the average current. They are equal to 39.9% and 15.3% for pristine CNT and CNT/NiO, respectively. The decrease in emission current fluctuations due to the coating of NiO nanotubes may be due to the lower susceptibility of tips to ion bombardment in the residual atmosphere, when uncoated tubes experience it more strongly, which can lead to their sputtering. A similar phenomenon was observed by the authors of [26]. Thus, according to the experimental data, the introduction of an oxide coating on the tubes showed a slight deterioration in the emission characteristics. However, another result was shown in [9], in which NiO formed an inhomogeneous coating from closely spaced crystal clusters, demonstrating an increase in the field emission current of nanotubes. Such a difference in the results could be due to the difference in coating morphologies, since in our case the ALD method of oxide coating was used, which gives a continuous uniform coating of tubes, and in [9] Ni nanoparticles deposited on an array of CNTs were oxidized. Probably, the effect of strengthening the focusing of the field on NiO particles decorating carbon nanotubes took place here.

## 4. Discussion

Among the reasons that led to the deterioration of the emission characteristics may be a geometric factor: an increase in the radius of the tips, due to which the focusing of the electric field decreased. In addition, among the possible reasons may be an increase in the work function of NiO to a value greater than that of nanotubes. It is noted in the literature that this value can be approximately 5.3 eV [22]. The appearance of an additional energy barrier for electrons at the point of contact between the oxide and the tube cannot be ruled out. In addition, among the reasons for the decrease in currents, one should consider a decrease in the electric field on the surface of the tubes passing through the oxide, which possibly has a sufficiently high concentration of free charge carriers in its structure, which endows NiO with the ability to screen the external electric field, which leads to its decrease when passing to the tubes. To more accurately establish the mechanism for reducing emission currents, additional studies of the electronic structure, conductivity, concentration, and type of charge carriers of NiO films obtained by the ALD method should be carried out with the same technological growth parameters.

Obviously, the uniformity of the current output over the cathode area is low. To achieve greater uniformity, it is necessary to manufacture structures with a smaller spread of tips in height. In addition, for a more detailed analysis of the effect of different NiO thicknesses, it is necessary to prepare a batch of samples with the same geometric characteristics and separate tips, which is a technically difficult task. Carrying out such studies is beyond the scope of this work, since it is only in the nature of preliminary tests. However, this type of array shows a good result in terms of overall current output and current stability at low operating voltages. This type of cathode with short tubes is a suitable basis for a comparative analysis of the effects on field emission of various coatings and a promising candidate for the manufacture of electron sources in various technical applications.

## 5. Conclusions

In this work, we studied the influence of NiO on the field emission characteristics of CNT arrays in terms of key parameters over the entire operating range. Analysis of the data showed that the deposition of an oxide coating on CNTs slightly worsened their emission characteristics. The measured IVCs for pristine CNT and CNT/NiO showed that turn-on fields were 6.16 V/µm for clean nanotubes and 9.66 V/µm for 7.5 nm thick NiO-coated CNTs. An increase in E_th_ from 4.85 V/µm to 7.49 V/µm was also found upon coating with oxide. The maximum recorded emission current densities were 2.86 mA/cm^2^ and 0.86 mA/cm^2^ for clean and coated tubes, respectively. This could be caused either by an increase in the radius of the tips, due to which the focusing of the electric field decreased, by an increase in the work function to a value greater than that of nanotubes, or by a decrease in the electric field on the surface of the tubes passing through the oxide, which possibly has a rather high concentration in its structure of free charge carriers, which give NiO the ability to screen the external electric field, which leads to its decrease when passing to the tubes. To more accurately establish the mechanism for reducing emission currents, additional studies of the electronic structure, conductivity, charge concentration, and type of charge carriers should be carried out.

However, the deposition of an oxide coating favorably affected the emission current fluctuations. They are equal to 40% and 15% for pristine CNT and CNT/NiO, respectively. Improving the emission stability due to the coating of nanotubes with NiO may be due to the lower susceptibility of tips to ion bombardment in the residual atmosphere.

## Figures and Tables

**Figure 1 nanomaterials-12-03463-f001:**
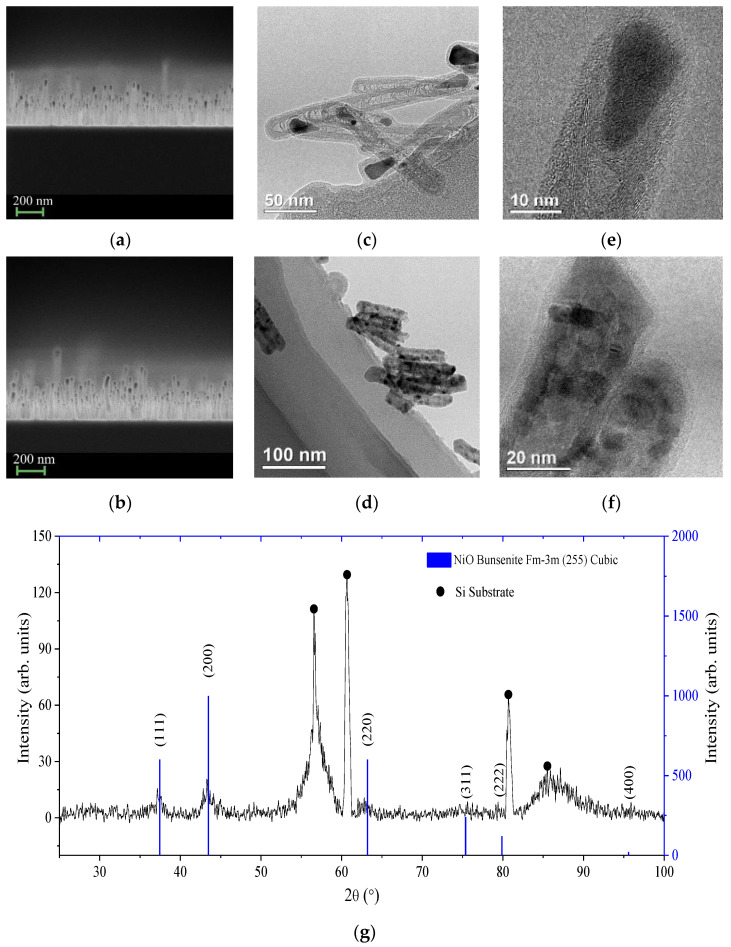
SEM images for pristine (**a**) and NiO-coated (**b**) CNTs. TEM images for pristine (**c**) and NiO-coated (**d**) CNTs, detailed TEM image of pristine CNT (**e**), detailed TEM image of NiO-coated CNTs (**f**,**g**) and XRD spectrum of NiO coated CNTs.

**Figure 2 nanomaterials-12-03463-f002:**
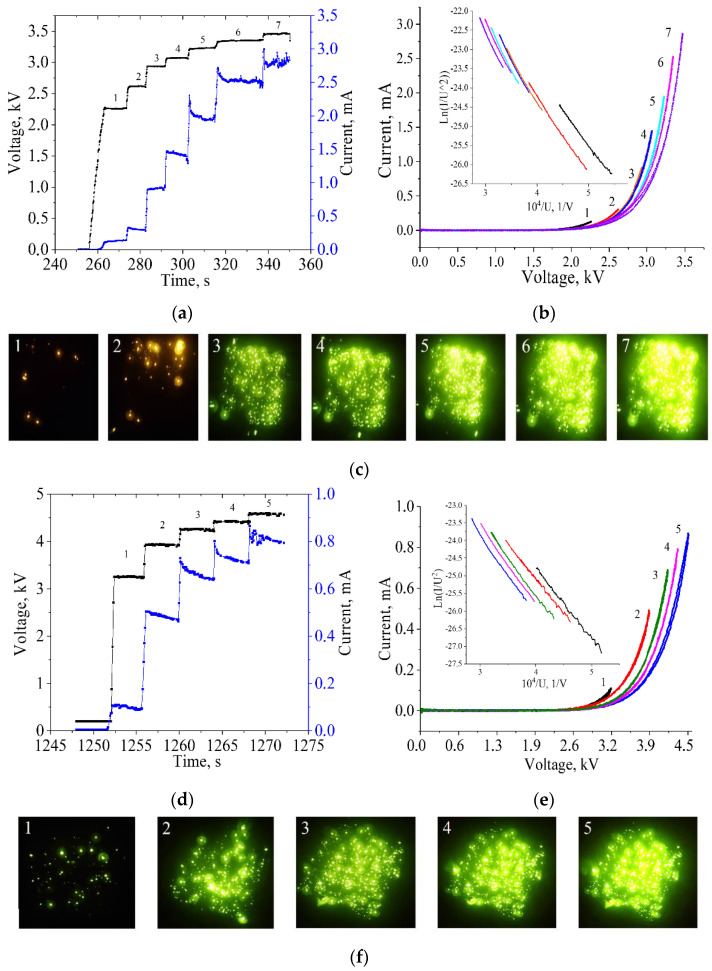
Training of cathodes by voltage steps for pristine CNT (**a**) and CNT/NiO (**d**), IVCs for samples (corresponding to the step numbers and glow patterns (**c**,**f**)) and corresponding IVC in Fowler−Nordheim coordinates (in inserts) for pristine CNT (**b**) and CNT/NiO (**e**).

**Figure 3 nanomaterials-12-03463-f003:**
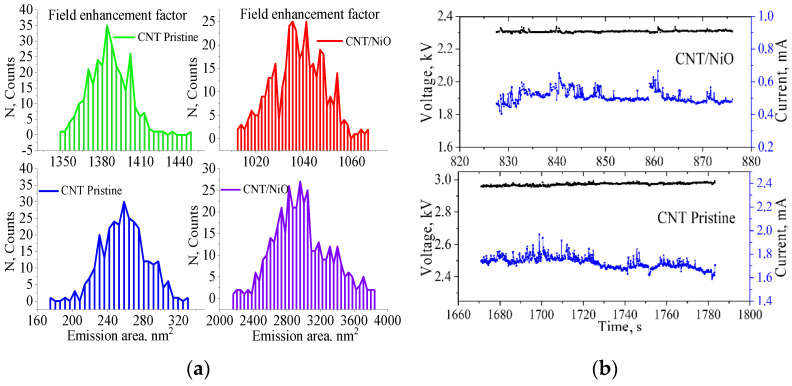
Statistics of the effective field enhancement factor and emission area (**a**) and the current fluctuations of the samples (**b**).

## Data Availability

All data generated or analyzed during this study are available within the article.

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
