# Peer review of "Influence of NiO ALD Coatings on the Field Emission Characteristic of CNT Arrays"

_nanomaterials, 2022, doi:10.3390/nano12193463_

Round 1

Reviewer 1 Report

1.       The paper studied the effect of NiO coating on the field emission from CNTs. The emission current and turn-on field was not improved by the NiO coating. I would suggest the author to vary the coating thickness and see the effect of the coating.

2.       Author claimed the stability of the NiO coated CNT was improved. However, from the data presented in the paper “15.3 % and 39.9 % for pristine CNT and CNT/NiO, respectively.(Page 6)”, the NiO coated CNTs have poorer stability.

3.       Furthermore, the definition of stability equation (1) is not proper. The equation actually gives the fluctuation, i.e. instability.

4.       In the SEM pictures in Fig. 1 the magnification for the CNT and CNT/NiO samples are different. Please use the picture in the same magnification for comparation.

Reviewer 2 Report

The authors performed an experimental study on NiO coated CNT, results are properly discussed and explained. However, the authors need to better demonstrate the significance of this study that worth publication, and better distinguish this study to others (especially ref 11). Below are my detailed comments.

  1. The emission characteristics of Ref 11 was improved with NiO coating, but these characteristics are worsen in this study. I agree with the authors that this maybe caused by thicker NiO coating, leading to larger tip radius. But couldn’t the authors use a thiner NiO coated CNT to carry this experiment? 
    1. If the emission characteristic worsen, then what’s the significance of this study? I assume the stability was also better with NiO coating in ref 11. 
  2. How’s the chemical composition of NiO coating affect the emission characteristic? If the author varies Ni : O atomic ratio, would it leads to a better work function match of NiO and CNT, thus improves the emission characteristics? 
  3. For fig 2 c and f, what are the emission center spatially located? It will be good to show a side be side SEM or TEM image.
  4.  

Reviewer 3 Report

In this manuscript, the authors examine the effect of a conformal NiO layer on the emission characteristics of vertically aligned carbon nanotubes. 

I have three comments for the authors to consider:

1. It seems to me that the authors were initially trying to improve the emission characteristics through conformal NiO coatings following a previous study (ref. 11). 

However, the authors are seeing a completely opposite behavior to that reported in ref. 11, which I think requires an explanation in the context of the differences in the samples used in ref. 11 and those used in the present study.

2. The authors mention that little information is available regarding the effect of NiO composition, structural, and electronic features on the emission performance (line 44), which I thought to be the motivation of this study.

However, none of the mentioned features were investigated in this study and I feel that the statement is very misleading.

3. The authors claim that there's a large variation in the height of vertically aligned carbon nanotubes. This is one of the critical factors that determine the emission characteristics and I am concerned that the characteristics reported in figure 2 may not necessarily be representative of all prepared samples.

Round 2

Reviewer 1 Report

The authors have answered the questions.

Reviewer 2 Report

The authors did not even try to address any of the questions, so I assume the questions can be considered as technical flaws of this work. As a result, I would like to suggest a "Reject" to this manuscript. 

Reviewer 3 Report

The authors have addressed my concerns and therefore I am in favor of publication.
